# Atorvastatin inhibits Lipopolysaccharide (LPS)-induced vascular inflammation to protect endothelium by inducing Heme Oxygenase-1 (HO-1) expression

Jian Luo[1]☯, Qian Zhu[1]☯, Keming Huang[1], Xue Wen[2], Yongquan Peng[1], Gong Chen📷[1]*, Gang Wei[1]*

1 Department of Cardiology, The Affiliated Hospital of Southwest Medical University, Lu Zhou, China,
2 Department of Laboratory Medicine, The Affiliated Hospital of Southwest Medical University, Lu Zhou, China

☯ These authors contributed equally to this work.
* chengong198692@swmu.edu.cn (GC); wg553725703@swmu.edu.cn (GW)

**Data Availability Statement:** All relevant data are within the manuscript and its Supporting Information files.

## Abstract

### Purpose

This study aimed to explore the differential effects of varying doses of atorvastatin on antagonizing lipopolysaccharide (LPS)-induced endothelial inflammation based on heme oxygenase 1 (HO-1) expression.

### Method

Vascular endothelial inflammatory injury was induced in 40 Sprague-Dawley rats by intraperitoneal injection of LPS. These rats were randomly divided into control, low-dose atorvastatin, high-dose atorvastatin, and HO-1 blocking groups. Seven days after treatment, all rats were sacrificed, and heart-derived peripheral blood was collected to measure the serum concentrations of bilirubin, alanine aminotransferase (ALT), total cholesterol, malondialdehyde, endothelial cell protein C receptor, endothelin-1, von Willebrand factor, and soluble thrombomodulin. Meanwhile, the number of circulating endothelial cells was determined using flow cytometry. Vascular tissues from descending aorta of rats from each group were extracted to detect the expression level of HO-1.

### Results

After different doses of atorvastatin intervention, the above inflammatory indices were decreased, and HO-1 expression and ALT concentration were increased in the atorvastatin-treated group of rats compared with the control group. These changes were more pronounced in the high-dose statin group (P < 0.05). Conversely, no significant decrease in the above inflammatory indices and no significant increase in HO-1 expression were observed in rats in the blocking group (P > 0.05).

**Funding:** The author(s) received no specific funding for this work.

**Competing interests:** The authors have declared that no competing interests exist.

**Abbreviations:** LPS, lipopolysaccharide; HO-1, heme oxygenase 1; ZnPPIX, zinc protoporphyrin IX; ALT, alanine aminotransferase; TC, total cholesterol; ET-1, endothelin; MDA, malondialdehyde; EPCR, endothelial cell protein C receptor; vWF, von Willebrand factor; sTM, soluble thrombomodulin.

## Conclusion

For LPS-induced vascular inflammation, high-dose atorvastatin exerts potent anti-inflammatory and vascular endothelial protection effects by inducing HO-1 expression.

## 1 Introduction

Vascular inflammatory injury is characterized by inflammatory cell buildup in the vessel wall and endothelial dysfunction. This can potentially occur in any organ and lead to organ dysfunction. Vascular endothelial damage resulting from inflammation leads to cellular dysfunction, which is considered an early event in subsequent vessel wall diseases [1]. Cardiovascular disease is frequently accompanied by inflammatory damage to blood vessels, and endothelial inflammation is largely believed to be the origin of atherosclerosis [2]. Therefore, it is crucial to detect the early onset of endothelial cell injury with reliable markers and intervene early to restore normal endothelial function and prevent complications of vascular endothelial injury.

Lipopolysaccharide (LPS), a chemical compound found solely in the cell wall of Gram-negative bacteria, initiates a vascular inflammatory response upon entering the host bloodstream. This leads to increased secretion of endothelin-1 (ET-1) from cells and a noticeable increase in inflammatory factors, including malondialdehyde (MDA), endothelial cell protein C receptor (EPCR), and other cell damage markers, which implies impaired endothelial cell function [3,4]. When chronic inflammation of blood vessels occurs, endothelial nitric oxide synthase (NOS) is inhibited, leading to the reduced synthesis of nitric oxide (NO), a vasodilator [5,6]. This reduction is accompanied by increased reactive oxygen species (ROS) production by the enzyme nicotinamide adenine dinucleotide phosphate oxidase [2]. Consequently, an oxidative stress response occurs due to the accumulation of ROS, leading to vasodilatory disorder and endothelial cell dysfunction. Heme oxygenase 1 (HO-1) can respond to the pro-oxidant state generated by ROS by breaking down heme [7]. This leads to the production of ferrous ions, bilirubin, and carbon monoxide (CO), which subsequently exert antioxidant and anti-inflammatory effects [8,9].

A 3-Hydroxy-3-methylglutaryl coenzyme-A (HMG-CoA) reductase is a rate-limiting enzyme in cholesterol synthesis. Its inhibition by atorvastatin reduces cholesterol synthesis and the effects of adverse cardiovascular events. Further research has indicated that atorvastatin has anti-inflammatory properties, safeguards vascular endothelial cells, and preserves endothelial stability [10]. The antagonistic inflammatory effect of atorvastatin is beneficial in preventing and treating atherosclerotic diseases [11]. It has been demonstrated that atorvastatin possesses anti-inflammatory effects, which could be associated with HO-1 expression [12]. After entering cells, statins can exert anti-inflammatory effects by activating mitogen-activated protein kinase (MAPK), anti-apoptotic kinase, and nuclear factor erythroid 2-related factor 2 while inhibiting the nuclear factor-kappa B (NF-κB) signaling pathway to increase HO-1 expression [13].

Regarding lipid regulation, the cardiovascular benefits of cholesterol reduction with high-dose statins compared to conventional dosing have been recognized. Regarding anti-oxidative stress effects, previous studies have shown that statins have anti-inflammatory effects, although at doses exceeding the established standard [14]. However, when high-dose statins were used in follow-up studies to treat inflammation caused by photochemical damage, the observed anti-inflammatory effects were ineffective [15]. The anti-inflammatory effects of statins showed some variability in the inflammation induced by different causative factors. Consequently, this study focused on identifying the differences in the anti-inflammatory effects of

different doses of atorvastatin on LPS-induced vascular inflammation and the possible mechanisms underlying its anti-inflammatory effects.

## 2 Materials and methods

### 2.1 Animals and materials

The experimental animals were 40 healthy 8-week-old Sprague-Dawley rats with an average weight of 180–200 g, purchased from the Medical Animal Center of Southwest Medical University. All experimental animals were approved by the Animal Ethics Committee of the Affiliated Hospital of Southwest Medical University.

The research materials purchased for this study included atorvastatin (Institute for Standard Drug Control, Lu Zhou, China), zinc protoporphyrin IX (ZnPPIX; Sigma, St. Louis, Missouri, USA), LPS (Solarbio, Beijing, China), ELISAs for MDA, ET-1, von Willebrand factor (vWF), and soluble thrombomodulin (sTM), EPCR kit (Abcam, Cambridge, UK), HO-1 rabbit polyclonal antibody (Abcam, Cambridge, UK), and cluster of differentiation 31 (CD31) flow cytometry antibody (Becton, Dickinson and Company, New Jersey, USA).

### 2.2 Methods

**2.2.1 Grouping and treatment.** Forty healthy 8-week-old Sprague-Dawley rats (half male and half female) were intraperitoneally injected with LPS (5 mg/kg/day (d)) for 15 days to induce vascular endothelial inflammatory damage. Subsequently, the rats were mixed and randomly divided into four groups (n = 10 each): control (vehicle), low-dose atorvastatin, high-dose atorvastatin, and HO-1 blocking groups. Four groups of rats were treated with 0.9% normal saline at 10 mL/d, atorvastatin at 2 mg/kg/d, atorvastatin at 10 mg/kg/d, and atorvastatin at 10 mg/kg/d + ZnPPIX at 5 mg/kg/d via intraperitoneal injection. Seven days after treatment, all rats were sacrificed, and cardiac blood samples were collected to analyze serum bilirubin, alanine aminotransferase (ALT), total cholesterol (TC), MDA, EPCR, NO, ET-1, vWF, and sTM concentrations. The number of circulating endothelial cells (CECs) was determined by flow cytometry. The descending aortas of rats in each group were harvested to determine HO-1 expression in the vascular tissue.

This experiment was approved by the Animal Ethics Committee of the Affiliated Hospital of Southwest Medical University.

**2.2.2 Determination of serum bilirubin, ALT, and TC concentrations.** Cardiac blood samples of rats were analyzed using an automated biochemical analyzer to determine the serum bilirubin, ALT, and TC concentrations.

**2.2.3 Determination of serum MDA, ET-1, vWF, sTM, and EPCR.** Serum concentrations of MDA, ET-1, vWF, sTM, and EPCR were determined using rat cardiac blood samples according to the instructions of each ELISA kit.

**2.2.4 Determination of peripheral CECs.** One milliliter of rat cardiac blood sample was collected in a 1 mL centrifuge tube and lysed using a blood cell lysate reagent. Subsequently, 1 μL of CD31 antibody was added to each centrifuge tube, incubated in the dark for 30 min, washed twice with 500 μL phosphate buffer saline (PBS) buffer, centrifuged at 1200 rpm for 5 min, and resuspended in 300 μL PBS for flow cytometry detection. The number of CECs is positively correlated with the number of CD31 positive cells.

**2.2.5 Detection of HO-1 expression in vascular tissues.** Tissue samples (1–2 cm) were collected from the proximal descending aorta of the rats and washed with pre-cooled PBS buffer at 4°C to remove residual blood, fascia, fat, and other redundant tissues. After weighing, these tissues were added to centrifuge tubes and minced, followed by adding PBS buffer at a 1:5 ratio. The mixture was slowly poured into a pre-cooled tissue homogenizer until no tissue clumps were

visible and the color was uniform. The resulting tissue homogenate (3 mL) was centrifuged at 3000 rpm for 10 min to separate the microsomes, and the protein concentration was quantified using the Coomassie Brilliant Blue method. Under the microscope, cytoplasm and nuclei with light yellow or brown staining were considered positive for HO-1 expression. Ten 400× high-power fields were selected to count positive cells. The average value was taken as a single sample.

## 2.3 Statistical analysis

Data analysis and statistical plotting were performed using Statistical Package for the Social Sciences (SPSS; version 26.0) and GraphPad Prism (version 9.3.0) software. The normally data results were expressed as mean ± SD. One-way ANOVA was used for comparing means across more than three groups, while comparisons of means between two groups were conducted using the Student's t-test to determine the statistical significance with a P < 0.05 considered statistically significant, 95% CI was used to provide confidence in the estimations.

## 3 Results

### 3.1 Effect of atorvastatin on serum biochemical parameters

Seven days after atorvastatin treatment in rats with inflammation, we determined several bio-chemical indice,including serum bilirubin, ALT and cholesterol (S1 and S3 Tables). The results of the study found that serum bilirubin concentration was elevated in rats in the small-dose statin-fed group compared to the control group (7.04±0.69 vs 7.94±0.60, 95% CI: -1.73 to -1.36, P = 0.028), and there was also a significant elevation in serum bilirubin concentration in the high-dose statin-fed group (7.04±0.69 vs 8.88±0.78, 95% CI: -2.67 to -1.01, P<0.0001). The elevation of serum bilirubin was more pronounced in the high-dose statin group than in the low-dose group (7.94±0.60 vs 8.88±0.78, 95% CI: -1.77 to -0.11, P = 0.021) (Fig 1A).The results also found that the serum concentration of ALT was elevated in both the high-dose statin-fed (96.60±6.02 vs 127.91±8.83, 95% CI: -40.43 to -22.2, P<0.0001) and the low-dose statin-fed (96.60±6.02 vs 108.91±7.56, 95% CI: -21.43 to -3.19, P = 0.0046) groups compared with the control group, and the elevation of ALT was more pronounced in the high-dose statin group than in the low-dose group(127.91±8.83 vs 108.91±7.56, 95% CI: 9.89 to 28.12, P<0.0001) (Fig 1B).However, the serum cholesterol concentrations in rats across the three atorvastatin-treated groups were not significantly different from those of the control group (P = 0.386; Fig 1C).

### 3.2 Atorvastatin treatment reduces the serum inflammatory indices

In experimental rats with LPS-induced vascular inflammation, serum inflammatory indices were tested following treatment with varying doses of atorvastatin (S2 and S4 Tables). The serological concentrations of MDA, ET-1, vWF, sTM, and EPCR,which reflect the degree of inflammation, roughly follow the following trend(Fig 2). Compared with the control group, the serum inflammatory indices of rats were reduced to different degrees after treatment with both low and high doses of statins (P < 0.05). Notably, the levels of these inflammatory indices were even lower in the high-dose statin-treated group of rats than in the low-dose group (P < 0.05). However, following ZnPPIX addition, no significant difference was observed in these inflammatory indices between the high-dose and control groups (P < 0.05).

### 3.3 Atorvastatin reduces the number of serum CECs in vascular inflammatory rats

After treatment with different doses of atorvastatin, we determined the proportion of CD31-positive cells in the serum of different rat groups (S5 Table). The proportion of

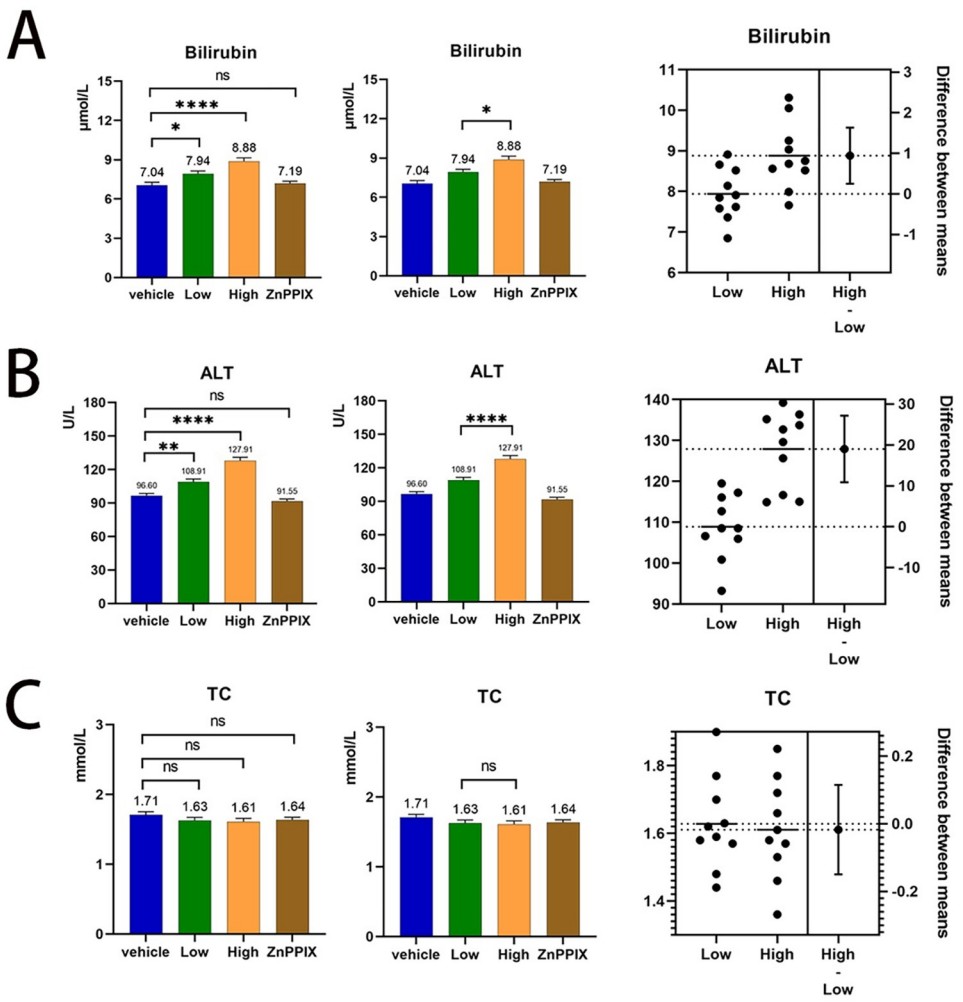

**Fig 1. Comparison of serum bilirubin, ALT and cholesterol in four groups.**

CD31-positive cells was positively correlated with the number of CECs. We found a significantly lower proportion of CD31-positive cells in both the high-dose(21.01±0.57 vs 12.61 ±0.75, 95% CI: 6.88 to 9.92, P<0.0001) and low-dose (21.01±0.57 vs 17.02±0.89, 95% CI: 2.47 to 5.51, P<0.0001) statin groups than in the control grou. Moreover, the high-dose statin group showed a more significant reduction than the low-dose statin group (12.61±0.75 vs 17.02±0.89, 95% CI: -5.93 to -2.89, P<0.0001). Conversely, the HO-1-blocked group exhibited a significant increase(30.31±2.17 vs 21.01±0.57, 95% CI: 7.78 to 10.82, P<0.0001) in CECs compared to the control group (Fig 3).

### 3.4 Atorvastatin increases HO-1 expression in rat vascular tissue

After administering varied doses of atorvastatin, we measured the HO-1 levels in the vascular tissue of the proximal segment of the descending aorta in rats (S6 Table). The HO-1 expression was increased and intensified upon staining in both the high-dose statin (OD value of HO-1 by Coomassie Brilliant Blue method,the same as below, 14730±518 vs 68160±1715, 95% CI: -54717 to -52143, P<0.0001) and low-dose statin(14730±518 vs 26350±775, 95% CI: -12907 to -10333, P<0.0001) groups compared to that in the control group. Quantification using the

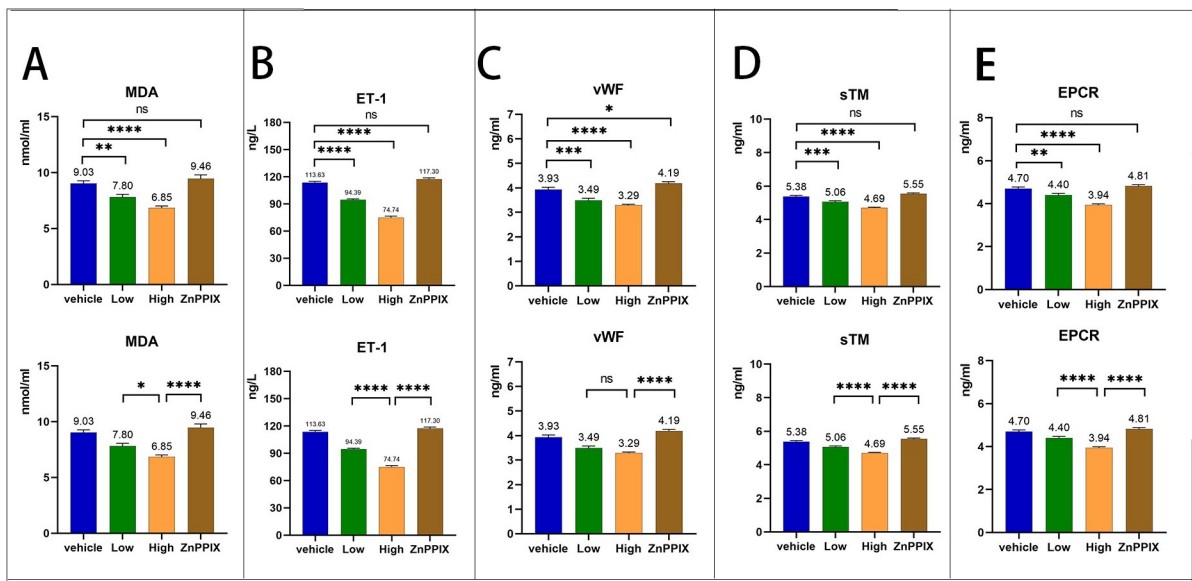

**Fig 2. Comparison of serum inflammatory markers in rats in four groups.**

assay revealed a statistically significant difference, with the high-dose atorvastatin group displaying a larger and deeper stained area than the low-dose atorvastatin group(26350±775 vs 68160±1715, 95% CI: -43097 to -40523, P<0.0001) (Fig 4).When compared to the control group, HO-1 expression was reduced in the HO-1 blocking group.(14730±518 vs 10160±548, 95% CI: 3283 to 5857, P<0.0001).

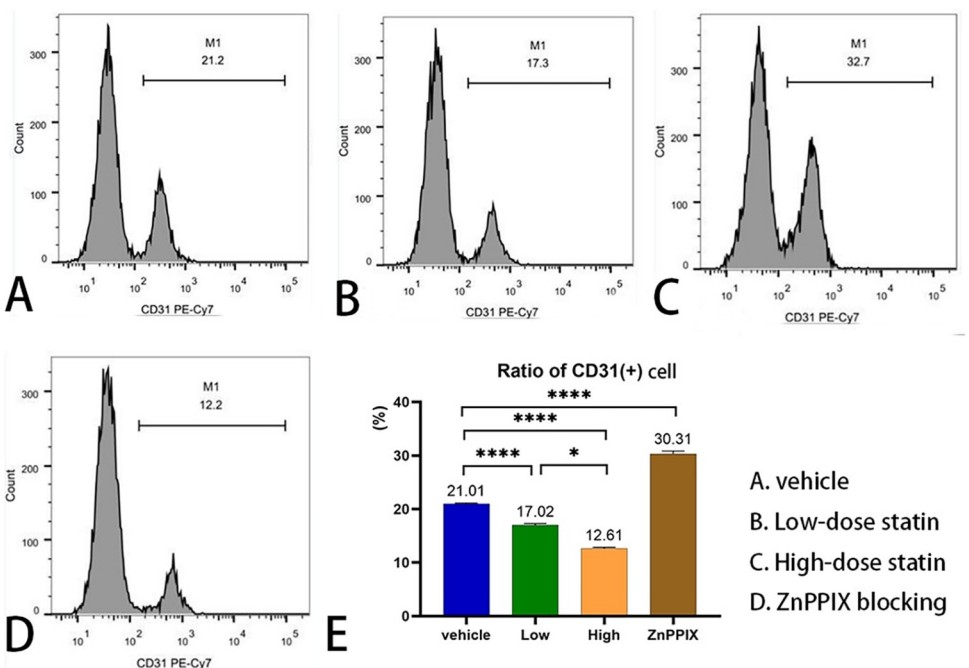

**Fig 3. Proportion of serum CD31 positive cells in four groups.**

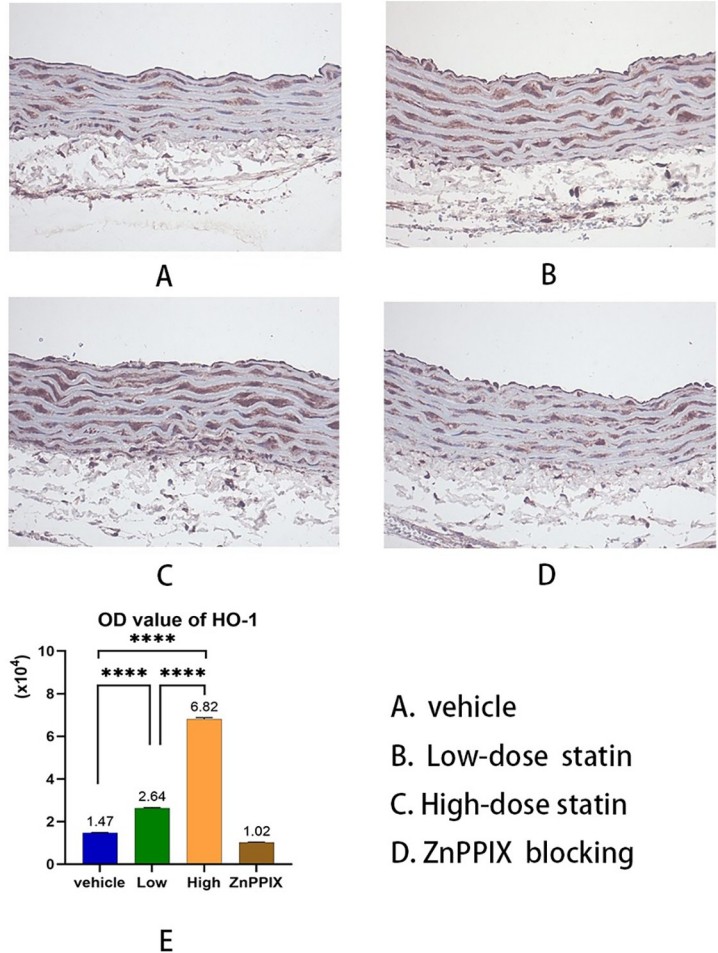

**Fig 4. HO-1 expression in descending aortic vascular tissue.**

## 4 Discussion

Vascular inflammation triggers endothelial damage, leading to narrowing or occlusion of the vascular lumen and even aneurysm or excessive bleeding upon severe vessel wall damag [16]. Besides being an effective cholesterol-lowering agent, atorvastatin also has antioxidant, anti-inflammatory, and vascular endothelial protective effects. It induces HO-1 expression and breaks down heme, an important component of the endogenous anti-oxidative stress system [17]. The present study found a significant increase in serum bilirubin concentration in high-dose atorvastatin-treated rats. Additionally, the high-dose atorvastatin-treated group showed a significant reduction in serological inflammatory indicators, including MDA and ET-1, compared to the low-dose group. These findings indicated that high-dose atorvastatin treatment may be more effective in reducing LPS-induced vascular inflammation than low-dose atorvastatin treatment.

Moreover, our results on serum CEC levels support the above findings. Circulating endothelial cells undergo necrosis or apoptosis due to physical strain, enzymatic free radicals, cytokine protease medications, or other factors that affect the vascular endothelium [18]. These inactivated cells subsequently detach from the basement membrane and enter the bloodstream. They are dependable indicators of vascular injury, with their numbers significantly

increasing in disorders characterized by extensive vascular damage [19]. Studies have shown that determining CECs can predict the severity and prognosis of various vascular-related diseases like Kawasaki disease, sepsis, and hypertension [20–22]. In our study, the serum CEC count in rats with vascular inflammation was significantly reduced after atorvastatin treatment compared to that in the control group. Conversely, a more pronounced reduction was observed in the high-dose statin group. These results support the conclusion that atorvastatin exhibits an inhibitory effect on vascular inflammation and protects endothelial cells, and that its inhibitory effect on vascular inflammation is more pronounced in high-dose atorvastatin.

The enzyme heme oxygenase (HO) is critical to heme metabolism. Its isoenzymes, HO-1 and HO-2, break down heme into biliverdin, CO, and ferrous iron [23]. Biliverdin is eventually reduced to bilirubin in mammals. The isoenzyme HO-1 acts as a metabolic enzyme for hemoglobin conversion and a potentially impactful multipurpose mediator in the inflammatory process, thereby contributing to inflammatory regulation [24]. The antioxidants biliverdin and bilirubin, along with CO that activates the p38 MAPK signaling pathway, and ferrous ions involved in ferritin synthesis, constitute pathways that exert anti-oxidative stress [9,25,26]. The HO-1 plays a crucial role in resisting oxidative stress injury. However, this anti-oxidative function can be obstructed by ZnPPIX, a metal ion-exchange derivative of protoporphyrin [27]. In our study, the high-dose atorvastatin-treated rat group exhibited a reduction in serological markers, reflecting inflammation and a decreased count of CECs. Conversely, no such changes were observed in the blocked group that received high-dose atorvastatin along with ZnPPIX. Analysis of HO-1 expression in the descending aortic endothelium revealed significantly lower HO-1 levels in the ZnPPIX-added blockade group than in the high-dose statin group alone. Consequently, we believe that the anti-oxidative stress and anti-inflammatory effects of atorvastatin may be associated with the stimulation of HO-1 expression.

Although the high-dose atorvastatin administration showed improved anti-inflammatory effects, the corresponding group of rats showed increased levels of transaminases, implying that the potent anti-inflammatory properties of high-dose atorvastatin are accompanied by a certain degree of organ dysfunction. Based on this, we believe that LPS-induced vascular inflammation can be reduced by monitoring vascular endothelial function and risk stratification. By optimizing the atorvastatin dosage in different inflammation levels, we can achieve its anti-inflammatory effects while reducing its side effects.

Regarding serum TC in this study, atorvastatin reduced the cholesterol levels to varying degrees in each rat group. However, no significant difference was observed, possibly due to the short one-week atorvastatin administration period and the slow enzymatic reaction. Therefore, the lipid-lowering effect might not have reached the required time threshold to observe notable differences.

Lipopolysaccharide is a major component of bacterial endotoxins, and our experiments using LPS to induce vascular inflammation and treating it with different atorvastatin doses revealed that the higher dose of atorvastatin showed more significant anti-inflammatory effects. Moreover, several studies have investigated the use of atorvastatin in combating bacterial and fungal infections [28,29]. Considering these investigations and our research findings, it is conceivable to hypothesize that short-term shock therapy with high-dose atorvastatin might reduce damage to the circulatory system caused by diseases related to vascular inflammation (especially endotoxin-induced vascular inflammation). However, this hypothesis requires further investigation.

In conclusion, this study revealed that high-dose atorvastatin exhibits effective anti-inflammatory properties and protects endothelial cells against LPS-induced vascular inflammation by inducing HO-1 expression.

## Supporting information

**S1 Table. Serum bilirubin, ALT and TC concentrations in each group.**
(DOCX)

**S2 Table. Concentrations of serum MDA, ET-1, NO, vWF, sTM, and EPCR in each group.**
(DOCX)

**S3 Table. Biochemical indicator values in each group.**
(XLSX)

**S4 Table. Inflamation indicator values in each group.**
(XLSX)

**S5 Table. CEC positive percentage in each group.**
(XLSX)

**S6 Table. OD value of HO-1 expression in each group.**
(XLSX)

## Acknowledgments

We express our heartfelt thanks to Southwest Medical University for providing us with laboratories, and also thank Home for Researchers editorial team for language editing service.

## Author Contributions

**Conceptualization:** Keming Huang, Xue Wen, Gang Wei.

**Data curation:** Jian Luo.

**Formal analysis:** Gang Wei.

**Funding acquisition:** Jian Luo, Gang Wei.

**Methodology:** Jian Luo.

**Project administration:** Qian Zhu, Yongquan Peng.

**Resources:** Keming Huang, Yongquan Peng.

**Software:** Keming Huang.

**Writing – original draft:** Gong Chen.

**Writing – review & editing:** Gong Chen.

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
