## [Decision Letter · Decision Letter 0]

2 Jun 2024

PONE-D-24-15138High-dose atorvastatin is significantly in mitigating LPS-induced vascular inflammation by inducing HO-1 expressionPLOS ONE

Dear Dr. Chen,

Thank you for submitting your manuscript to PLOS ONE. After careful consideration, we feel that it has merit but does not fully meet PLOS ONE’s publication criteria as it currently stands. Therefore, we invite you to submit a revised version of the manuscript that addresses the points raised during the review process.

Please submit your revised manuscript by Jul 17 2024 11:59PM. If you will need more time than this to complete your revisions, please reply to this message or contact the journal office at plosone@plos.org. Please include the following items when submitting your revised manuscript:A rebuttal letter that responds to each point raised by the academic editor and reviewer(s). You should upload this letter as a separate file labeled 'Response to Reviewers'.A marked-up copy of your manuscript that highlights changes made to the original version. You should upload this as a separate file labeled 'Revised Manuscript with Track Changes'.An unmarked version of your revised paper without tracked changes. You should upload this as a separate file labeled 'Manuscript'.

We look forward to receiving your revised manuscript.

Kind regards,

Misbahuddin Rafeeq

Academic Editor

PLOS ONE

Journal Requirements:

"no"

Reviewers' comments:

Reviewer's Responses to Questions

**Comments to the Author**

1. Is the manuscript technically sound, and do the data support the conclusions?

Reviewer #1: Yes

Reviewer #2: Yes

Reviewer #3: Yes

2. Has the statistical analysis been performed appropriately and rigorously? 

Reviewer #1: Yes

Reviewer #2: Yes

Reviewer #3: Yes

3. Have the authors made all data underlying the findings in their manuscript fully available?

Reviewer #1: Yes

Reviewer #2: Yes

Reviewer #3: No

4. Is the manuscript presented in an intelligible fashion and written in standard English?

Reviewer #1: Yes

Reviewer #2: Yes

Reviewer #3: Yes

5. Review Comments to the Author

Reviewer #1: Rephrase title

Page 8, results section, it ill be good to add a list of the indices investigated. ...'we determined several biochemical indices' such as ......

Page 10, discussion, just wondering the unique and the new thing the current research brought on board. This is because much work on atorvastatin in mitigating LPS-induce vascular inflammation had been extensively explored. Kindly address this...

Reviewer #2: First Round Report

• The authors discuss an important disease vascular inflammation that is diagnosed in many different diseases and suggest different dosages for atorvastatin to relieve the inflammation effect.

• The authors presented a clear introduction sequentially to illustrate the relationship between vascular inflammatory injury and how it is related to lipopolysaccharide (LPS) and dose-dependent anti-inflammatory effects of atorvastatin on LPS-induced vascular inflammation and its mechanisms.

“Vascular inflammatory injury and how it is related to lipopolysaccharide (LPS), found in the cell wall of Gram-negative bacteria, triggers a vascular inflammatory response, causing endothelial dysfunction characterized by increased secretion of endothelin-1 (ET-1) and cytokines. Chronic inflammation inhibits endothelial nitric oxide synthase (NOS), reducing nitric oxide (NO) synthesis and increasing reactive oxygen species (ROS) production, leading to oxidative stress and endothelial dysfunction. Heme oxygenase 1 (HO-1) responds to ROS by exerting antioxidant effects. Atorvastatin, known for its cholesterol-lowering effects, also exhibits anti-inflammatory properties by modulating various pathways, including HO-1 expression”.

• In the Materials and methods, I found that the abbreviations mentioned as follows: “MDA, EPCR, NO, ET-1, vWF, and sTM concentrations” are unclear to me. So, I suggest adding the abbreviation after the abstract to easily guide the readers through the whole manuscript.

• In the results, the authors mentioned in the study design that the used rats were half male and half female, which was not shown in the description of the results, and as a result, the legends for the figures confused the readers about which one is referred to the male or the female.

• In the discussion, the authors address the outcomes clearly, but it is important to clarify the differences in the outcomes for males and females to avoid bias in showing and discussing the results.

Reviewer #3: 1. Is the manuscript technically sound, and do the data support the conclusion?

Yes

- Strength

- Technically it sounds good, the outcome is tested twice with CEC – circulating endothelial cells count and for tissue from direct descending aorta and both results are in coherence.

- For quantitative analysis of protein expression in the tissue the Coomassie Brilliant Blue method provides visual confirmation but may lack quantitative accuracy, but using high-power fields and high expertise could minimize the error. The ten 400x high power field where ten separate fields of view observed at a magnification of 400 times can ensure representative sampling of the tissue to increase the statistical robustness of the findings and it is enough as their tissue homogenate is homogenous.

- Typical sample size for experimental groups of Sprague-Dawley rats ranges from 8-12, of course it depends on statistical power, scientific validity and ethical consideration. The 10 sample size for each of four groups is enough sample size for this study. Controls are also used reverse control also being done by blocking HO-1 expression.

- The result, discussion and conclusion are inline and appropriate

- The biomarkers for endothelial injury and damage selected here (endothelial cell protein c receptors, endothelin-1, von Willbrand factor, and soluble thrombomodulin are appropriate, and others to measure the effect of atorvastatin such as HMG-CoA reductase for cholesterol TC, HO-1 expression and the LPS inflicted damage to the endothelial is also appropriate. The mechanism underlying the effect or atorvastatin through HMG-G inhibition and increasing HO-1 expression is also well explained.

- To be clarified or corrected

- The dosage of atorvastatin for low and high is 2 and 10 mg/kg/d. In researches involving Sprague-Dawley rats the dosage of atorvastatin; typically, or approximately range for low dose 1-5, medium 5-20 and high is above 20 mg/kg/day, here as the dose can vary based on specific research objective (here is to test high dose), duration of the study and weight of rats; it would be better if you mention the reason why you select the 2 and 10 mg dosage per kg/d, what is your background or could you reason out? Have you considered previous literature or do you have any dose-response studies to determine the most appropriate doses?

2. Has the statistical analysis been performed appropriately and rigorously?

Yes,

Strength

- one-way ANOVA and students t-test is used as their data was parametric, that is correct to compare the 3 or 2 groups, the right statistical methods used.

To be clarified or corrected

- Better to put the actual p value along with the cutting value 0.05 to get a highlight of how much it is significant in the result section instead of putting only like <0.05 or >0.05

- instead of saying the normally or approximately normally you should put the nature of distribution of your data aptly as normal or not or with the other words like parametric or non-parametric – this makes your analysis ambiguous so avoid the saying approximately here.

- In addition to this if multiple doses of atorvastatin are tested here, is it not possible to analyze dose-response relationships using regression analysis or non-linear modeling techniques– this is optional if you add this it will strengthen your study

- at what CI was the P value calculated it should have to be mentioned in the statistical methodology part at least once

3. Have the authors made all data underlying the findings in their manuscript fully available?

- No

- The average measurements were presented in the figures at the apex. The descriptive statistics part should be presented as additional data; the means, medians and variance measures should be available, according to PLOS data policy. It is better to be included here.

4. Is the manuscript presented in an intelligible fashion and written in standard English?

Yes

Strength

- The manuscript is presented with clear and standard English.

To be corrected or clarified

- The title “ ..is significantly in mitigating …“ is grammatically incorrect, the adjective significantly is used as a noun here. Use it as significant or if you prefer to use significantly you need to put it with its subordinate word such as high.– the third option is to restructure or change the word.

- In the introduction section “increased secretion of endothelin-1 (ET-1) from cells and a noticeable increase in cytokine levels, including malondialhdehyde (MDA), endothelial cell protein C receptor (EPCR), and ” in this sentence you should avoid including as it misleads that MDA is also a cytokine which it is not.

- In the abstract –“ based on heme oxygenase “ should be “ based on heme oxygenase expression”

- 1.3 statistical analysis

o “The normally or approximately normally distributed data” – this phrase has grammatical error instead of saying the normally you need to say the normal or approximately normal, this comment is from the language perspective but statistically you need to be apt, if it is parametric just say it is parametric and if not just put non-parametric .

o “meanSEM.One-way” – no space after full stop

o The confidence level should also be mentioned along with the p value results at least once

- 1.2.4

o “..CD31 positive cells.” Put the full stop.

- 2.2

o “following trend(Figure 2)” need to be following trend (Figure 2) - put the space

5. Review comments to the Author

- Generally, it is interesting to review a technically rich and thoroughly made study like this. The Authors has put great effort to make a significant contribution to the scientific community and to the existing body of knowledge. Technically the study is rich and the presentation also so much interesting. I am happy to read their manuscript to review it. It is well done and keep it up guys.

6. PLOS authors have the option to publish the peer review history of their article (what does this mean?). If published, this will include your full peer review and any attached files.

Reviewer #1: No

Reviewer #2: No

Reviewer #3: No

---

## [Author Response · Author response to Decision Letter 0]

30 Jun 2024

We have revised the article in accordance with the reviewers' and editors' comments.

Part A (Reviewer 1)

The reviewer's comment 1: Rephrase title.

The author's answer: We reformulated the title of the article according to the reviewer's request and the main idea of the article, and finally changed the original title "High-dose atorvastatin is significantly in mitigating LPS-induced vascular inflammation by inducing HO-1 expression" to "Atorvastatin Inhibits Lipopolysaccharide (LPS)-induced Vascular Inflammation to by inducing HO-1 expression". The new title emphasizes the role of atorvastatin in inhibiting inflammatory activity and protecting the vascular endothelium in inhibiting lipopolysaccharide (a major component of endotoxin)-induced inflammatory vascular injury and briefly describes the mechanism that may be related to HO-1 expression. The new article title echoes the main idea. In contrast to the previous title, the new title not only briefly summarizes the main points of the study, but also highlights the promising application of atorvastatin in the treatment of vascular inflammatory injury due to bacterial endotoxin infection. (page 1, lines 2-4).

The reviewer's comment 2: Page 8, results section, it ill be good to add a list of the indices investigated. ...'we determined several biochemical indices' such as ......

The author's answer: Thank you to the reviewers for their practical advice. Given that multiple biochemical metrics were measured in the article, we believed it was necessary to state in advance which metrics were measured before presenting the findings for each metric. Therefore, we have added the following at the beginning of the biochemical section of the experimental results as suggested by the reviewers:Seven days after atorvastatin treatment in rats with inflammation, we determined several biochemical indices, including serum bilirubin, ALT and cholesterol.(page 8, lines 4-5)

The reviewer's comment 3: Page 10, discussion, just wondering the unique and the new thing the current research brought on board. This is because much work on atorvastatin in mitigating LPS-induce vascular inflammation had been extensively explored. Kindly address this...

The author's answer: Thanks to the reviewers for their valuable feedback. Indeed, the use of atorvastatin for the treatment of LPS-associated vascular inflammation has been explored, and the mechanism may be related to the up-regulation of p38 MAPK, p13k/Akt, and nrf2 with the down-regulation of nf-κB. All of these signaling pathways ultimately promote HO-1 expression, which in turn exerts anti-inflammatory effects. As stated in the introductory section of the article, from the results of the current study, there is a relative lack of studies on the dose of atorvastatin and the strength of the anti-inflammatory effect, and the results of different studies are divergent (page 5, lines 2-5) . Therefore, while validating atorvastatin's antagonism of LPS-induced inflammation, our study focuses on whether atorvastatin has a dose-effect relationship in its anti-inflammatory effects as it is used for lipid-lowering and plaque stabilization in atherosclerotic lesions. Our study found that high-dose atorvastatin showed a more superior anti-inflammatory effect, suggesting that intensive statin therapy may have some promising applications in suppressing LPS-induced inflammation (page 13, lines 11-14).

Part B (Reviewer 2)

The reviewer's comment 1: In the Materials and methods, I found that the abbreviations mentioned as follows: “MDA, EPCR, NO, ET-1, vWF, and sTM concentrations” are unclear to me. So, I suggest adding the abbreviation after the abstract to easily guide the readers through the whole manuscript.

The author's answer 1: Thank you to the reviewers for their suggestions, indeed, our study involves many indicators, and they appear in the article in abbreviated form in the later text, except for the first time they appear when the full name is explained, which may cause distress to the readers who need to consult the text again. Therefore, we considered it necessary to add the abbreviations of these recurring indicators. Following the reviewers' comments, we have added the abbreviations of the recurring metrics after the abstract of the article in the order in which they appear for quick reference by readers. Thank you again. (page 3, lines 1-4)

The reviewer's comment 2: In the results, the authors mentioned in the study design that the used rats were half male and half female, which was not shown in the description of the results, and as a result, the legends for the figures confused the readers about which one is referred to the male or the female.

The author's answer 2: We thank the reviewers for the comments. In fact, in our study, the experimental rats were divided equally between males and females, and all the rats were mixed before experimental grouping, and the randomization was carried out among the rats that had already been mixed. The purpose of this was to minimize the effect of gender on the experimental results. Therefore, the final results are not gender-specific. To avoid misunderstanding, we have explained the grouping in the Materials and Methods section of the article in more detail by adding "Subsequently, the rats were divided into four groups..." to the article and correcting it to read "The rats were divided into four groups...". We added the text "Subsequently, the rats were mixed and randomly divided into four groups..." to correct it to "Subsequently, the rats were mixed and randomly divided into four groups...". (page 6, line 3)

The reviewer's comment 3: In the discussion, the authors address the outcomes clearly, but it is important to clarify the differences in the outcomes for males and females to avoid bias in showing and discussing the results.

The author's answer 3: We apologize for any misunderstanding that may have been caused by the reviewers due to our misrepresentation. As stated in the previous question, we have mixed all rats before experimental grouping, which means that there are both male and female rats in each final experimental group, and the experimental results are based on the whole experimental group population. Therefore, our experimental results do not clarify the differences between the results of magnetic and male rats.Exploring whether gender has any effect on the anti-inflammatory effects of statins would be interesting research. However,it is not the scope of this current work,we would love to investigate it in the future.Thank you.

Part C (Reviewer 3)

The reviewer's comment : The dosage of atorvastatin for low and high is 2 and 10 mg/kg/d. In researches involving Sprague-Dawley rats the dosage of atorvastatin; typically, or approximately range for low dose 1-5, medium 5-20 and high is above 20 mg/kg/day, here as the dose can vary based on specific research objective (here is to test high dose), duration of the study and weight of rats; it would be better if you mention the reason why you select the 2 and 10 mg dosage per kg/d, what is your background or could you reason out? Have you considered previous literature or do you have any dose-response studies to determine the most appropriate doses?

The author's answer: Thanks to the reviewers for this feedback. Indeed, in the pharmacologic study of atorvastatin, our review of the literature revealed some differences in the determination of the drug dose from study to study. We determined the dose selection for the rat study in the following way: In general, for a standard weight human (body weight of 60 kg, body surface area of 1.62 m2), atorvastatin 80 mg/d is the upper limit of the recommended dose determined by the current study, and the use of the drug beyond this dose will greatly increase the incidence of adverse effects of statin. Due to the species differences between animals and humans, it is inappropriate to apply the human statin dose to rats after directly converting it to body weight. We therefore calculated the equivalent dose for rats based on the Human Equivalent Dose (HED), which is usually calculated for animals based on the clinical dose for humans. The human dose is converted to an animal dose using the body surface area conversion formula. This method takes into account metabolic differences between species.The HED can be calculated usually based on a conversion between body surface area or body weight, as metabolic rates, body surface areas, and body weight ratios vary between species. Typically, HED is calculated based on the conversion of Body Surface Area (BSA). [ HED , (mg/kg) = Animal , dose , (mg/kg)/times (Animal , Km / Human , Km) ]. Where Km value is the conversion factor for body surface area, and different species have different Km values. Based on the above formula, we can calculate the conversion factor for the drug dose from human to Sprague-Dawley rat as 6.17. Therefore, for a standard body size of 60 kg, based on the human dose of atorvastatin of 80 mg per day, we can calculate the human equivalent dose for the rat as (80 mg/60 kg) * 6.17 ≈ 10 mg(as weight within ±25% is acceptable). Similarly we can calculate the Similarly, it can be calculated that a small statin dosage of 2mg per day is reasonable. Reference: 1.FDA Guidance for industry and reviewers: Estimating the safe starting dose in clinical trials for therapeutics in adult healthy volunteers [S]. 2.FDA Guidance for Industry：food-effect bioavailability and fed bioequivalence studies［S］．

Part D (Editors)

The editors 's comment 1 :Better to put the actual p value along with the cutting value 0.05 to get a highlight of how much it is significant in the result section instead of putting only like <0.05 or >0.05.

The author's answer 1: Thanks for the editorial opinion. We have listed the specific p-values in the article article that involve statistical comparisons in the text as suggested, and we have also listed the mean ± standard deviation of the two sets of data for which comparisons were made, and the 95% confidence intervals for the difference in means.

The editors 's comment 2 : instead of saying the normally or approximately normally you should put the nature of distribution of your data aptly as normal or not or with the other words like parametric or non-parametric – this makes your analysis ambiguous so avoid the saying approximately here.

The author's answer 2: Thanks for the editorial comments. We believe that this comment is valuable and that ambiguity needs to be avoided in scientific research, so we have made changes in the statistical methods section of the article. After statistical calculations, our data were statistically calculated to fit a normal distribution, so we changed the original text from "The normally or approximately normally distributed data results were expressed...." to "The normally distributed data results were expressed...". Thanks again to the editors for your suggestion.(page 7,line 18)

The editors 's comment 2 :at what CI was the P value calculated it should have to be mentioned in the statistical methodology part at least once.

The author's answer 2: we thanks for the editorial feedback. In order to make the statistical content of the article more complete and credible, we have added the confidence intervals for the P-values in the statistics section of the article according to the revised comments by adding the following "95% CI was used to provide confidence in the estimations." ( Page 7, line 22)

The editors 's comment 3: The title “ ..is significantly in mitigating …“ is grammatically incorrect, the adjective significantly is used as a noun here. Use it as significant or if you prefer to use significantly you need to put it with its subordinate word such as high.– the third option is to restructure or change the word.

The author's answer 3: We are very grateful to the reviewers and editors for their valuable revisions to the text. Combining the comments of the reviewers1 and the editor, we have reformulated the title of the article, and the new title has been changed to "Atorvastatin Inhibits Lipopolysaccharide (LPS)-induced Vascular Inflammation to Protect Endothelium by Inducing Heme Oxygenase-1 (HO-1) Expression". The new title emphasizes the role of atorvastatin in inhibiting inflammatory activity and protecting the vascular endothelium in inhibiting lipopolysaccharide (LPS)-induced vascular inflammatory injury, a major component of endotoxin, and briefly describes the mechanism that may be related to HO-1 expression. Corrections are kindly requested if there are any inaccuracies. Thank you.

The editors 's comment 4: In the introduction section “increased secretion of endothelin-1 (ET-1) from cells and a noticeable increase in cytokine levels, including malondialhdehyde (MDA), endothelial cell protein C receptor (EPCR), and ” in this sentence you should avoid including as it misleads that MDA is also a cytokine which it is not.

The author's answer 4: Thanks to the editors for this feedback. We apologize for the oversight in writing the article .malondialhdehyde (MDA) is not belong to cytokine. We changed the article from cytokines to inflammatory factors when we revised the article so that MDA would be included (page 3, line 20)

The editors 's comment 5: In the abstract –“ based on heme oxygenase “ should be “ based on heme oxygenase expression”

The author's answer 5: Thanks to the editor's suggestion, we have revised the article according to the editor's suggestion, so that the revision makes the article more complete.(page 2, line 4)

The editors 's comment 6: meanSEM.One-way” – no space after full stop; 

The author's answer 6: Thanks to the editors for this feedback. We have made changes in the corresponding part of the article in response to the alerts.(page 7, line 19)

The editors 's comment 7: “..CD31 positive cells.” Put the full stop;

The author's answer 7: Thanks to the editors for this feedback. We have made changes in the corresponding part of the article in response to the alerts. (page 7, line 2)

The editors 's comment 8: “following trend(Figure 2)” need to be following trend (Figure 2) - put the space

The author's answer 8: Thanks to the editors for this feedback. We have made changes in the corresponding part of the article in response to the alerts. (page 10, line 7)

Other revisions made by the author at the request of the journal.

1. We have adjusted the position of the figures and tables in the article according to the magazine's instructions, placing them in the corresponding parts of the article.

2. We have adjusted the citation format of the references in the article to the "Vancouver" format.

---

## [Editor Report · Decision Letter 1]

31 Jul 2024

Atorvastatin Inhibits Lipopolysaccharide (LPS)-induced Vascular Inflammation to Protect Endothelium by Inducing Heme Oxygenase-1 (HO-1) Expression

PONE-D-24-15138R1

Dear Dr. Gong Chen,

We’re pleased to inform you that your manuscript has been judged scientifically suitable for publication and will be formally accepted for publication once it meets all outstanding technical requirements.

Kind regards,

Misbahuddin Rafeeq

Academic Editor

PLOS ONE

---

## [Editor Report · Acceptance letter]

8 Aug 2024

PONE-D-24-15138R1 

PLOS ONE

Dear Dr. Chen, 

I'm pleased to inform you that your manuscript has been deemed suitable for publication in PLOS ONE. Congratulations! Your manuscript is now being handed over to our production team.

Kind regards, 

on behalf of

Dr. Misbahuddin Rafeeq 

Academic Editor

PLOS ONE